# The Aqueous Extract of *Brassica oleracea* L. Exerts Phytotoxicity by Modulating H_2_O_2_ and O_2_^−^ Levels, Antioxidant Enzyme Activity and Phytohormone Levels

**DOI:** 10.3390/plants12173086

**Published:** 2023-08-28

**Authors:** Yu Wang, Yuanzheng Zhao, Baozhu Dong, Dong Wang, Jianxiu Hao, Xinyu Jia, Yuxi Zhao, Yin Nian, Hongyou Zhou

**Affiliations:** 1College of Horticulture and Plant Protection, Inner Mongolia Agricultural University, Hohhot 010020, China; wang_style@emails.imau.edu.cn (Y.W.);; 2Institute of Plant Protection, Inner Mongolia Academy of Agricultural & Husbandry Sciences, Hohhot 010031, China; 3State Key Laboratory of Phytochemistry and Plant Resources in West China, Kunming Institute of Botany, Chinese Academy of Sciences, Kunming 650201, China

**Keywords:** allelopathy, reactive oxygen species, antioxidant enzymes, phytohormones, LC-MS

## Abstract

Allelopathic interactions between plants serve as powerful tools for weed control. Despite the increasing understanding of the allelopathic mechanisms between different plant species, the inhibitory effects of *B. oleracea* on weed growth remain poorly understood. In this study, we conducted experiments to demonstrate that *B. oleracea* extract can suppress the germination of *Panicum miliaceum* L.varruderale Kit. seeds as well as of the roots, shoots and hypocotyl elongation of *P. miliaceum* seedlings. Furthermore, we observed that *B. oleracea* extract reduced the levels of hydrogen peroxide and superoxide anion in the roots while increasing the activities of catalase and ascorbate peroxidase. In the shoots, *B. oleracea* extract enhanced the activities of superoxide dismutase and peroxidase. Moreover, the use of the extract led to an increase in the content of phytohormones (indole-3-acetic acid, indole-3-acetaldehyde, methyl indole-3-acetate, N6-isoPentenyladenosine, dihydrozeatin-7-glucoside, abscisic acid and abscisic acid glucose ester) in *P. miliaceum* seedlings. Interestingly, the aqueous extract contained auxins and their analogs, which inhibited the germination and growth of *P. miliaceum*. This may contribute to the mechanism of the *B. oleracea*-extract-induced suppression of *P. miliaceum* growth.

## 1. Introduction

Weeds pose a significant challenge to crop production, as they compete with crops for essential resources such as soil nutrients and water [1,2]. To address this issue, chemical herbicides have been widely used and have introduced significant benefits to agriculture [3]. However, these herbicides also come with several disadvantages, including the development of weed resistance [4,5]. Consequently, there is an urgent need to improve weed control methods in a more sustainable manner. One promising approach is allelopathy, a biological phenomenon involving chemical interactions between plants [6,7]. Allelopathy has the potential to be an effective and environmentally friendly tool for the weed management of field crops [8]. By harnessing the natural chemical compounds produced by certain plants, it is possible to suppress the growth and development of weeds [9,10].

The Brassicaceae family holds significant potential for the exertion of allelopathic effects between weeds and other crops [11]. Incorporating the residues of *B. oleracea*, a Brassicaceae vegetable, into the soil can significantly reduce weed biomass and weed species diversity [12]. However, the specific effects of *B. oleracea* on the physiochemistry of weeds are not yet fully understood. Reactive oxygen species (ROS) play crucial roles in various biological processes within plant growth [13,14,15]. Variations in the accumulation of superoxide anion (O_2_^−^) and hydrogen peroxide (H_2_O_2_) in root tips have a significant impact on root growth and differentiation [16,17,18].

Antioxidant systems in cell membranes and other organelles are closely related to ROS and help prevent oxidative damage by regulating the production and elimination of ROS [19]. This includes antioxidant defense enzymes such as superoxide dismutase (SOD), ascorbate peroxidase (APX), peroxidase (POD) and catalase (CAT) [20]. Further investigation is needed to understand the role of the antioxidant enzyme system in the exertion of allelopathic effects.

Allelochemicals can modulate plant growth and development by influencing phytohormones and cell division [21]. Some allelochemicals can reduce ethylene and auxin levels in plants, thereby inhibiting the growth and development of recipient plants [22]. Understanding the physiological and biochemical mechanisms involved, such as the role of ROS and antioxidant defense enzymes, as well as the modulation of phytohormones by allelochemicals, is crucial for comprehending the full extent of allelopathic effects.

Currently, limited knowledge is available regarding the effects of allelochemicals released by *B. oleracea* on the physiological and biochemical aspects of *P. miliaceum*, a grass weed species. Therefore, this study investigates the influence of *B. oleracea* on the growth parameters (shoot length, root length, hypocotyl length and germination rate), antioxidant enzyme activities (CAT, APX, POD and SOD) and phytohormone levels (auxins, cytokinins and abscisic acid) of *P. miliaceum*. Additionally, the components of the *B. oleracea* extracts are identified. These findings may provide valuable insights into the weed control of *B. oleracea* and offer useful parameters for its management.

## 2. Results

### 2.1. The Phytotoxicity of Aqueous Extract on P. miliaceum

During the observation of *P. miliaceum* seed germination, a strong inhibitory effect was noted as a result of an aqueous extract concentration of 0.28 g/mL, resulting in an 85% inhibition rate. Upon the addition of the aqueous extract, both the shoot and root growth of *P. miliaceum* seedlings were significantly suppressed, and the inhibitory effect increased with increasing concentrations of the extract. The strongest inhibition was observed on the roots when the concentration reached 0.28 g/mL. By calculating the EC_50_, it was found that the roots were the most sensitive to the aqueous extract, with an EC_50_ value of 0.08013 g/mL, indicating the strongest inhibitory effect (Figure 1).

### 2.2. The Effect of Aqueous Extract on H_2_O_2_ and O_2_^−^ Content in P. miliaceum Seedlings

The aqueous extract treatment resulted in a reduction in H_2_O_2_ and O_2_^−^ content in the roots of *P. miliaceum* seedlings (Figure 2a,b). The H_2_O_2_ content in the roots of *P. miliaceum* treated with the aqueous extract decreased to 2.65 ± 0.03 μmol/mL. Similarly, the O_2_^−^ content in the roots of the seedlings treated with the aqueous extract decreased significantly to 0.28 ± 0.04 μmol/mL compared to that of the control. However, there were no significant changes in the levels of H_2_O_2_ and O_2_^−^ in the shoots (Figure 2c,d).

### 2.3. Effect of Aqueous Extract on Antioxidant Enzyme Activity

Under the influence of the aqueous extract, the brightness of the CAT and APX bands isolated from the shoots of *P. miliaceum* on native gels was greater than that of the control. The activities of CAT and APX were enhanced in the shoots. In the roots, the brightness of CAT and PPO bands on the gel was higher than that of the control, indicating an increase in their activities. The activity of PPO showed different changes in the shoots and roots under the influence of the aqueous extract. However, the aqueous extract did not affect the activity of SOD (Figure 3).

### 2.4. The Effect of Aqueous Extract on Phytohormone Levels in P. miliaceum Seedlings

The treatment with the aqueous extract of *B. oleracea* resulted in significant increases in the contents of several phytohormones. The content of indole-3-carboxylic acid (ICA) increased by 34.8 ng/g. Additionally, there were significant increases in the contents of indole-3-carboxaldehyde (ICAID) and methyl indole-3-acetate (ME-IAA), with changes of 290.67 ng/g and 7.08 ng/g, respectively. Among the cytokinins, only the contents of dihydrozeatin-7-glucoside (DHZ7G) and N6-isoPentenyladenosine (IPR) showed significant increases. Moreover, the contents of abscisic acid (ABA) and abscisic acid glucosyl ester (ABA-GE) also increased significantly under the treatment with *B. oleracea* extract, with contents of 6.97 ng/g and 35.99 ng/g, respectively (Figure 4).

### 2.5. The Effects of Auxins on the Growth of P. miliaceum

We conducted experiments using ICA, ICAID and ME-IAA on the growth and development of *P. miliaceum* and calculated the EC_50_ values. We found that indole compounds had significant effects on the growth and development of *P. miliaceum*. In terms of the seed germination rate, ICAID showed the strongest inhibitory effect. For shoot elongation, ME-IAA had the lowest EC_50_ value, indicating the strongest inhibitory effect on shoots. Regarding root elongation, ICAID exhibited the most potent inhibitory effect (Figure 5).

### 2.6. Composition Analysis of B. oleracea Extract

After conducting a comparative analysis of the characteristic ion peaks using mass spectrometry, indole alkaloids were identified in the aqueous extract of *B. oleracea*. Among these, indole-3-acetic acid acts as an auxin, while indole-3-aldehyde, indole-3-acetonitrile and indole-3-carboxylic acid serve as analogs of auxin. Furthermore, the aqueous extract also contained other indole alkaloids, such as indole-3-acrylic acid and methoxyindoleacetic acid (Table 1). After we demonstrated the inhibitory effect of ICA on *P. miliaceum*, we also found the presence of ICA in the extract, indicating that ICA was one of the crucial molecules responsible for the phytotoxicity induced by *B. oleracea* extract.

## 3. Discussion

This study indicated that the aqueous extract obtained from *B. oleracea* inhibited the growth of *P. miliaceum*. The roots of *P. miliaceum* seedlings were the most sensitive to the aqueous extract, experiencing the strongest inhibitory effect. This was achieved through multiple mechanisms, including a reduction in reactive oxygen species (ROS) levels, an increase in the activity of antioxidant defense enzymes, and the modulation of phytohormone levels. ROS play a vital role as signaling molecules in regulating root growth [23,24]. Studies have demonstrated that spermidine and brassinosteroids can reduce the overall growth of maize seedlings by increasing oxidative damage through ROS production [25]. Additionally, the copper-mediated inhibition of primary root elongation is believed to occur via increasing H_2_O_2_ levels [26]. These findings suggest that ROS may serve as common signaling molecules for root growth across different plant species. However, our research findings indicated that the aqueous extract reduced the levels of H_2_O_2_ and O_2_^−^. Therefore, we concluded that H_2_O_2_ and O_2_^−^, as signaling molecules, generated different responses under various stress conditions. The treatment with the aqueous extract resulted in increased activities of POD, SOD, CAT and APX, which collectively contributed to a reduction in ROS accumulation in the roots. By enhancing the activities of these antioxidant defense enzymes, the extract treatment effectively mitigated ROS levels in root tissues [27]. *B. oleracea* extract significantly enhances the activity of CAT and PPO in *P. miliaceum* roots. However, in the shoots, treatment with *B. oleracea* extract altered the activity of CAT and APX, indicating that the activity of PPO and APX is organ-specific. Nevertheless, CAT activity was increased in both the roots and shoots, leading us to believe that CAT played a significant role in the phytotoxicity exerted by *B. oleracea* extract. *Artemisia absinthium* and *Psidium guajava* extracts increase the activity of CAT and APX in *Parthenium hysterophorus* leaves [28]. Nevertheless, further research is needed to understand the specific interplay between ROS clearance and antioxidant enzymes in allelopathic interactions.

Phytohormones play a crucial role in regulating seedling growth [29]. In this study, the content of the methyl ester of indole-3-acetic acid (ME-IAA) increased under treatment with the aqueous extract compared to that in the control. ME-IAA is an inactivated form of indole-3-acetic acid (IAA), while ME-IAA is believed to serve as a storage form that can be converted into IAA and utilized for homeostatic regulation in plants [30,31]. Meanwhile, our study also demonstrated that ME-IAA exerted a strong inhibitory effect on *P. miliaceum* growth. However, it neither exhibited a growth-promoting effect at low concentrations nor a growth-inhibitory effect at high concentrations, as is observed with auxin in plants [32,33]. Our research has revealed that *P. miliaceum* seedlings are influenced by the aqueous extract, which leads to an increase in the content of auxin. Interestingly, the aqueous extract also contains auxin. Exogenous auxin has been found to reduce H_2_O_2_ levels, thereby affecting the growth of plant seedlings. There exists an antagonistic relationship between H_2_O_2_ and auxin distribution. Therefore, we believe that the addition of exogenous auxin from *B. oleracea* results in the absorption of excessive auxin by *P. miliaceum*, leading to an antagonistic interaction with reactive oxygen species in the roots, thereby co-regulating the growth of *P. miliaceum* seedlings. However, the proportions and content of the phytohormones in *B. oleracea* extract need further determination, since these also accelerated the application of *B. oleracea* extract in field weed management.

Both ICA and ICAID are natural products, and ICAID has been demonstrated to play a significant role in the apical dominance of pea seedlings by acting as an inhibitor of lateral shoot growth. In addition, ICAID, as an inhibitor of TIR1, has promising prospects for development as a herbicide, [34,35], including during ethylene and ABA production in germinating seeds under allelopathic stress [36]. ABI4 is an AP2/ERF-type transcription factor that regulates the ABA signaling pathway during seed germination and post-germination growth [37]. ABI4 regulates the expression of genes involved in ROS production and the response to ROS clearance [38]. We believe that the increase in the content of abscisic acid activates the antioxidant system in *P. miliaceum* seedlings, thereby reducing the level of ROS. For example, ABI4 responds to salt stress in *Arabidopsis* by regulating ascorbate synthesis and ROS accumulation [39]. As a precursor of ABA, ABA-GE levels increase with elevated levels of ABA. Previous studies have suggested that the increase in endogenous ABA observed in plants may be partially attributed to the breakdown of endogenous ABA-GE [40,41].

Allelochemicals are bioactive mediators of allelopathy, and they are non-nutritional compounds formed as secondary metabolites during various physiological processes in plants [42]. Allelopathic effects typically arise from the combined action of different compounds. There is no single plant toxin exclusively responsible for exerting allelopathic effects [43]. Therefore, the phytotoxicity of *B. oleracea* extract on *P. miliaceum* was the result of the combined action of multiple allelochemicals.

In summary, *B. oleracea* exerts phytotoxic effects on *P. miliaceum* by reducing the germination of seeds, and inhibits root, shoot and embryo growth in *P. miliaceum* seedlings. Specific biochemical parameters associated with this allelopathic effect include a decrease in H_2_O_2_ and O_2_^−^ levels in the roots, as well as the induction of CAT and POD activity in the roots. Additionally, *B. oleracea* extract contains auxin and its analogs. This plant extract, which has the potential to control and manage weeds, warrants further research. If *B. oleracea* extract contains a variety of auxin analogs in abundant quantities and is used to regulate weed growth, it could contribute to weed management.

## 4. Materials and Methods

### 4.1. Preparation of the Aqueous Extract of B. oleracea

Briefly, 280 g of fresh *B. oleracea* leaves was weighed and ground into a powder using liquid nitrogen in a mortar. The powder was transferred into a 1 L conical flask, to which 1000 mL of sterile water was added. The flask was shaken with a shaker at 4 °C for 48 h. The resulting solution was filtered through four layers of cheesecloth and the filtrate was centrifuged at 4000 rpm for 20 min [44]. The supernatant was collected and a mother solution with a concentration of 0.28 g/mL was obtained. The mother solution was diluted with sterile water to obtain extraction solutions with concentrations of 0.01 g/mL, 0.03 g/mL, 0.05 g/mL, 0.07 g/mL, 0.14 g/mL and 0.28 g/mL. The prepared extraction solutions were then stored in a refrigerator at 4 °C for future use.

### 4.2. Germination and Growth Conditions of P. miliaceum

The growth inhibition of *P. miliaceum* was assessed using a medium composed of MS (2.16 g/L Murashige and Skoog basal medium, 8 g/L sucrose and 8 g/L agar). When the MS medium had nearly solidified, different concentrations of the sterilized extract were added, while the control group received an equal volume of sterile water. After sterilization, the *P. miliaceum* seeds were added to the culture bottles. *P. miliaceum* was cultivated at 23 °C under long day conditions (16/8 h light/dark) with cool-white fluorescence bulbs serving as the light source. The aqueous extract of *B. oleracea* was added to the medium before culturing the *P. miliaceum* [45]. The medium of the control treatment contained the same amount of water. After seven days, the germination rate, root length, shoot length and hypocotyl length of the *P. miliaceum* seeds were measured.

### 4.3. Determination of H_2_O_2_ and O_2_^−^ Levels

*P. miliaceum* seedlings were placed in a staining solution mixed with 10 mM Tris-HCl (pH 7.6) in an amount of 9 mL + DAB (3, 3-diaminobenzidine) + 2 mL of MES and vacuumed with a vacuum pump to allow the dye to enter the plant. They were stained in the dark for 24 h and then discolored with 95% alcohol and photographed; 0.5 g of NBT was dissolved in a 0.1 M phosphoric acid buffer to prepare 10 mL of the solution [46]. The ROS content was measured using assay kits (Suzhou Grace Biotechnology Co., Ltd.; Suzhou, China).

### 4.4. Assay of Antioxidant Enzyme Activity through Native-PAGE Profiling

Non-denatured polyacrylamide gel electrophoresis was performed on a vertical slab at 4 °C using 0.01 M Tris-glycine (pH 8.3) as an electrode buffer with 12% running and 3.5% stacking gel. Using a Mini-PROTEAN^®^ Tetra electrophoresis tank (Bio-Rad Laboratories Co., Ltd., Hercules, CA, USA) for electrophoresis. An enzyme sample corresponding to 30 mg of protein mixed with glycerol was layered on top of the stacking gel and electrophoresis was performed at a current of 25 mA per slab. After electrophoresis, the gels were washed with distilled water. The gel was incubated in 10 mM potassium phosphate buffer (pH 6.0) containing 20 mM guaiacol and 0.01% H_2_O_2_ for 15 min, and the POD isozymes were observed as orange-brown bands. After incubating them in 2.5 mmol/L NBT for 30 min, the SOD isozymes were observed on the gels as colored bands. After immersing the gels in 1.17 × 10^−6^ mol/L riboflavin for 20 min, they were transferred to Petri dishes and irradiated with fluorescent lamps. With the exception of the areas where SOD was localized, light exposure resulted in the production of a purple tint of insoluble formazan throughout the gel. The gels were incubated at room temperature in 0.1 M potassium phosphate buffer (pH 6.4) containing 4 mM ascorbate and 4 mM H_2_O_2_ for 15 min to localize the APX isozymes. Then, the gels were washed with distilled water and stained with 0.125 N HCl solution containing 0.1% ferrocyanide and 0.1% ferrichloride (*w*/*v*). Colorless bands against a Prussian blue backdrop were used to identify the APX isozymes. The CAT isozymes were visualized by soaking the gels in potassium phosphate buffer (pH 7.0) followed by incubating them in the same buffer containing 5 mM H_2_O_2_. After incubation for 10 min, the gels were rinsed with water and stained in a reactive mixture containing 2% (*w*/*v*) potassium ferricyanide and 2% ferric chloride (*v*/*v*). The isozymes were shown as colorless bands against a deep-blue background [47,48].

### 4.5. The Detection of Phytohormone Levels in P. miliaceum

*P. miliaceum* samples were ground to a powder using liquid nitrogen and extracted with 1 mL of methanol for subsequent LC-MS (AB 6500 + QTRAP^®^, Agilent Technologies Co., Ltd., Beijing, China) analysis. The liquid chromatography conditions were as follows: the column used was from Waters ACQUITY UPLC HSS T3 C18 (1.8 µm, 100 mm × 2.1 mm i.d.); the mobile phase consisted of water and acetonitrile; the quantification of phytohormones was performed based on the retention time and peak shape of the standard compounds; each peak area represented the relative content of the corresponding plant hormone; standard solutions with concentrations of 0.01 ng/mL, 0.05 ng/mL, 0.1 ng/mL, 0.5 ng/mL, 1 ng/mL, 5 ng/mL, 10 ng/mL, 50 ng/mL, 100 ng/mL and 200 ng/mL were prepared; the concentration versus peak area standard curve was calculated using the standard solutions; the peak areas of the *P. miliaceum* samples were then substituted into the standard curve equation to calculate the absolute content of phytohormones [49].

### 4.6. Bioassay of Auxin Analogs on P. miliaceum Germination and Growth

Aqueous solutions of indole-3-carboxaldehyde, indole-3-carboxylic acid and methyl indole-3-acetate with concentrations of 1 × 10^−1^ M, 1 × 10^−2^ M, 1 × 10^−3^ M, 1 × 10^−4^ M and 1 × 10^−5^ M were prepared. The compounds were dissolved via ultrasonic (SB5200DTD, Ningbo Scientz Biotechnology Co., Ltd., Ningbo, China) heating. A biological assay was performed using the Petri dish method. The germination rate of *P. miliaceum* seeds was calculated and the growth parameters of seedlings were measured.

### 4.7. Identification of Secondary Metabolites in Aqueous Extract of B. oleracea

The *B. oleracea* leaf extract was spun for 10 s to ensure thorough mixing. Briefly, 200 µL of the mixed sample was taken and transferred into a 2 mL EP tube. Then, 200 µL of 70% methanol was added and the sample was vortexed for 3 min before being centrifuged at 12,000 rpm at 4 °C for 10 min. The supernatant was filtered through an organic-based microporous membrane (0.22 μm) into a liquid phase injection vial and analyzed using LC-MS/MS. The liquid phase conditions were as follows: column, Agilent SB-C18 (1.8 µm, 2.1 mm × 100 mm); mobile phase A, ultrapure water (with 0.1% formic acid to ensure peak sharpness); mobile phase B, acetonitrile (0.1% formic acid); gradient elution, 0.00 min B at 5%, with an increase in the B proportion to 95% within 9.00 min, which was maintained for 1 min, and a decrease in the B proportion to 5%, from 10.00 to 11.10 min, which was equilibrated at 5% until the 14 min time point; flow rate, 0.35 mL/min; column temperature, 40 °C; injection volume, 4 μL. The main mass spectrometry conditions included the following ESI source operating parameters: ion source, turbo spray; source temperature, 550 °C; ion spray voltage (IS), 5500 V (positive ion mode)/–4500 V (negative ion mode); gas 1 (GS1), gas 2 (GS2) and curtain gas (CUR) set at 50, 60 and 25.0 psi, respectively; collision-induced dissociation (CID) parameters set to high. Instrument tuning and mass calibration were performed using 10 and 100 μmol/L of polyethylene glycol solutions for the QQQ and LIT modes, respectively. A QQQ scan was performed in the MRM mode with the collision gas (nitrogen) set to medium. The DP and CE for each MRM ion pair were optimized through further DP and CE optimization. A specific set of MRM ion pairs was monitored for each metabolite eluted during each period. The obtained mass spectrometry information was matched with that in the metabolite public database HMDB (http://www.hmdb.ca/ accessed on 20 June 2023) to obtain secondary metabolite information [50].

### 4.8. Statistical Analysis

The data were determined using the F-test or Levene’s test. Statistical significance was evaluated using a two-tailed *t*-test (for all two-group comparisons). The data are presented as mean ± standard error (SE) and a *p*-value of < 0.05 was considered statistically significant with * *p* < 0.05 and ** *p* < 0.01.

## 5. Conclusions

Our research on the allelopathic effect of *B. oleracea* on weeds indicates that *B. oleracea* aqueous extract can effectively inhibit the growth of *P. miliaceum*, making it a promising choice for sustainable weed management. Our study has revealed that the allelopathic activity of *B. oleracea* extract on *P. miliaceum* seedlings is mediated through the modulation of hydrogen peroxide, superoxide anions and phytohormone levels. Given the escalating issue of herbicide resistance, allelopathy has gained popularity due to its environmentally friendly nature. Therefore, our findings also demonstrate the potent phytotoxicity of *B. oleracea* extract on weeds, making it a viable approach for weed management.

## Figures and Tables

**Figure 1 plants-12-03086-f001:**
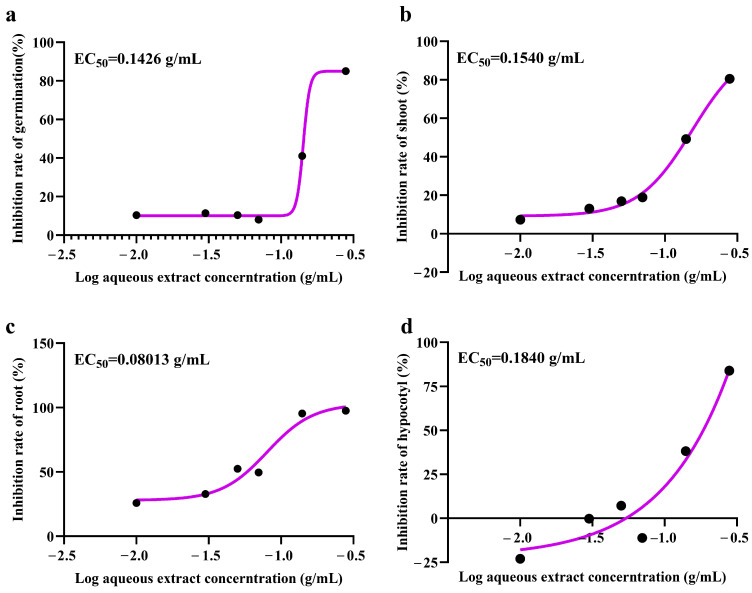
Effects of aqueous extract on the germination rate, root length, shoot length and hypocotyl length of *P. miliaceum*. (**a**) Effects of aqueous extract on the germination of *P. miliaceum*. (**b**) Effects of aqueous extract on the shoot of *P. miliaceum*. (**c**) Effects of aqueous extract on the root of *P. miliaceum*. (**d**) Effects of aqueous extract on the hypocotyl of *P. miliaceum*.

**Figure 2 plants-12-03086-f002:**
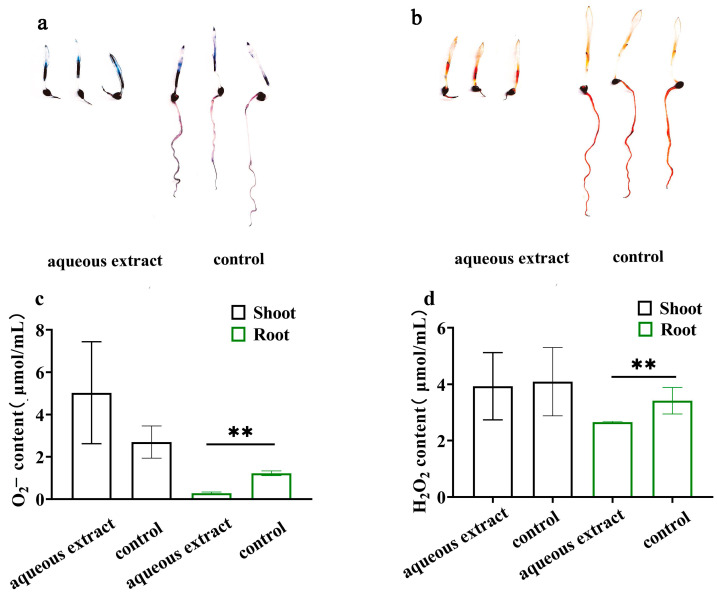
Effects of aqueous extract on H_2_O_2_ and O_2_^−^ accumulation in the roots and shoots of *P. miliaceum*. (**a**,**b**) Quantification of NBT and DAB staining. The blue staining represents O_2_^−^, while the red staining represents H_2_O_2_. (**c**,**d**) Content of O_2_^−^and H_2_O_2_ (** *p* < 0.01). Each bar graph is the mean ± standard error.

**Figure 3 plants-12-03086-f003:**
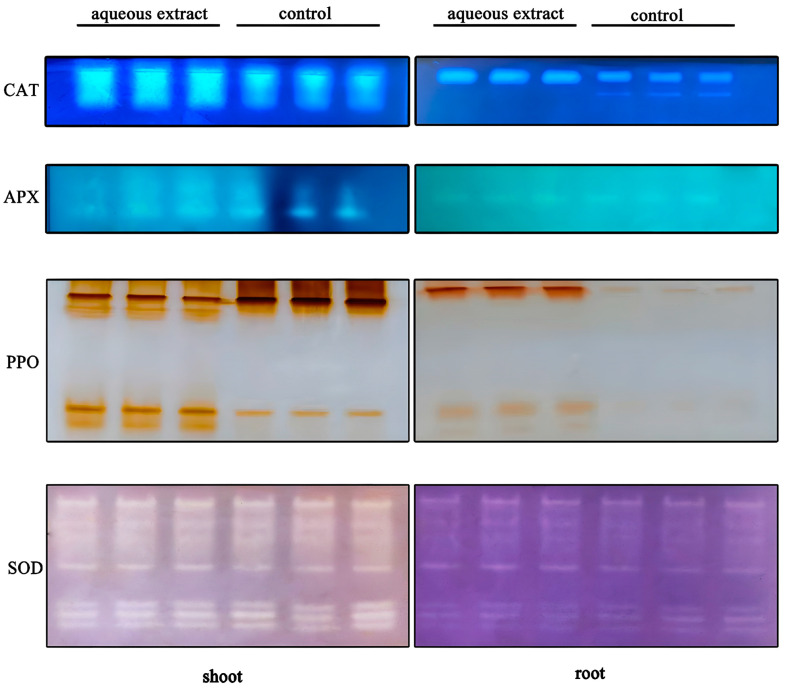
Isozyme profile of CAT, APX, PPO and SOD from the shoots and roots.

**Figure 4 plants-12-03086-f004:**
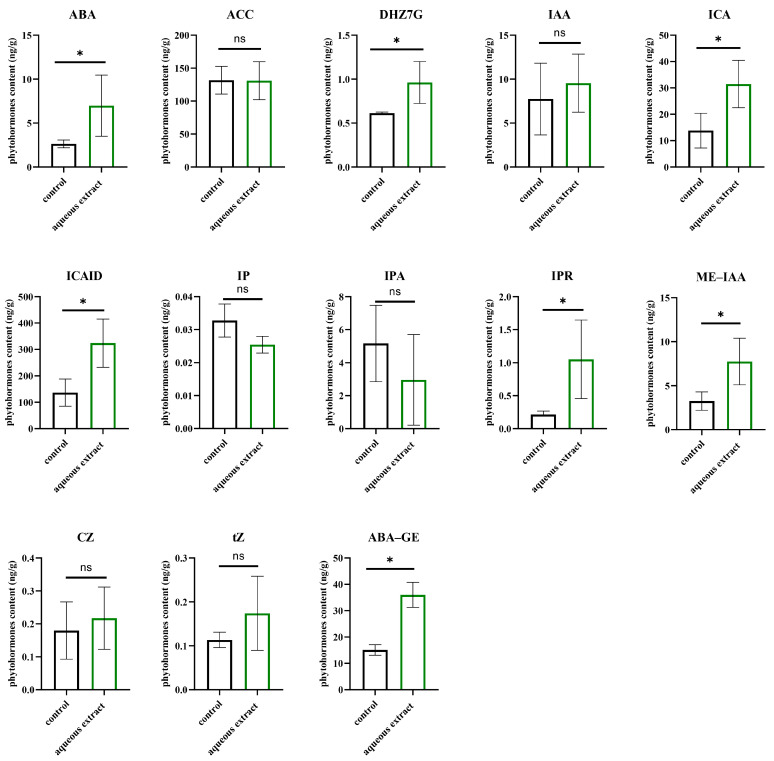
Phytohormone contents in aqueous extract compared to those in the control group. ABA: abscisic acid; ACC: 1-Aminocyclopropanecarboxylic acid; DHZ7G: dihydrozeatin-7-glucoside; IAA: indole-3-acetic acid; ICA: indole-3-carboxaldehyde; ICAID: indole-3-carboxylic Acid; IP: N6-isoPentenyladenine; IPA: 3-indolepropionic acid; IPR: N6-isoPentenyladenosine; ME-IAA: methyl indole-3-acetate; cZ: cis-Zeatin; tZ: trans-zeatin; ABA-GE: abscisic acid glucosyl ester. “ns” indicates that there was no significant difference after a *t*-test. (* *p* < 0.05).

**Figure 5 plants-12-03086-f005:**
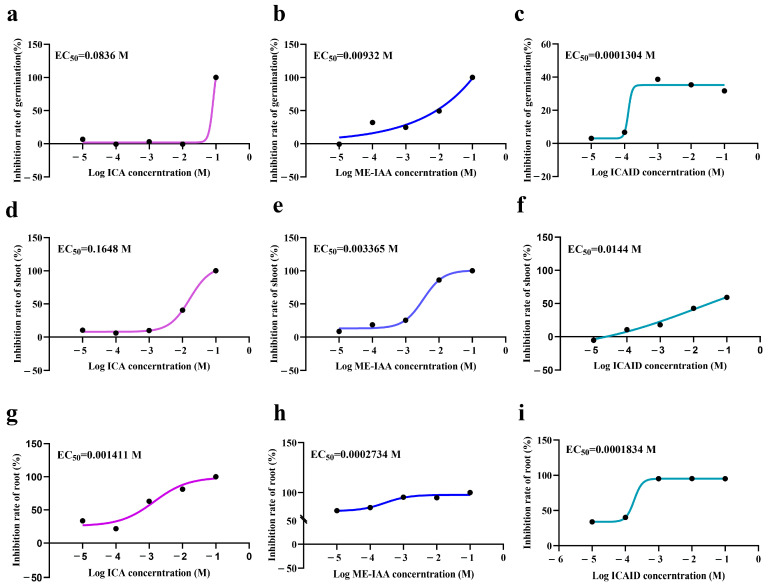
The effects of indole-3-carboxaldehyde (ICA), methyl indole-3-acetate (ME-IAA) and indole-3-carboxylic acid (ICAID) on the growth and development of *P. miliaceum*. (**a**) The effect of ICA on *P. miliaceum* germination. (**b**) The effect of ME-IAA on *P. miliaceum* germination. (**c**) The effect of ICAID on *P. miliaceum* germination. (**d**) The effect of ICA on *P. miliaceum* shoot length. (**e**) The effect of ME-IAA on *P. miliaceum* shoot length. (**f**) The effect of ICAID on *P. miliaceum* shoot length. (**g**) The effect of ICA on *P. miliaceum* root length. (**h**) The effect of ME-IAA on *P. miliaceum* root length. (**i**) The effect of ICAID on *P. miliaceum* root length.

**Table 1 plants-12-03086-t001:** Secondary metabolites in the aqueous extract of *B. oleracea*.

Formula	Ionization Model	Compounds
C_8_H_8_N_2_	[M+H]^+^	4-aminoindole
C_9_H_7_NO	[M+H]^+^	indole-3-carboxaldehyde
C_10_H_8_N_2_	[M+H]^+^	3-indoleacetonitrile
C_9_H_7_NO_2_	[M−H]^−^	indole-3-carboxylic acid
C_10_H_10_N_2_O	[M+H]^+^	3-indole acetamide
C_10_H_9_NO_2_	[M+H]^+^	1-methoxyindole-3-carbaldehyde
C_11_H_9_NO_2_	[M+H]^+^	3-indoleacrylic acid
C_11_H_11_NO_2_	[M+H]^+^	3-indolepropionic acid
C_11_H_12_N_2_O_2_	[M+H]^+^	1-methoxy-indole-3-acetamide
C_11_H_11_NO_3_	[M+H]^+^	methoxyindoleacetic acid
C_11_H_11_NO_3_	[M+H]^+^	3-hydroxy-3-acetonyloxindole
C_16_H_18_N_2_O_6_	[M−H]^−^	indole-3-cyano-6-o-glucoside
C_16_H_18_N_2_O_5_S	[M−H]^−^	indole-3-cyano-2-o-glucoside
C_10_H_8_N_2_	[M+H]^+^	3-indoleacetonitrile
C_10_H_10_N_2_O	[M+H]^+^	3-indole acetamide
C_10_H_9_NO_2_	[M+H]^+^	1-methoxyindole-3-carbaldehyde
C_11_H_9_NO_2_	[M+H]^+^	3-indoleacrylic acid
C_11_H_11_NO_2_	[M+H]^+^	3-indolepropionic acid
C_11_H_12_N_2_O_2_	[M+H]^+^	1-methoxy-indole-3-acetamide
C_11_H_11_NO_3_	[M+H]^+^	methoxyindoleacetic acid
C_11_H_11_NO_3_	[M+H]^+^	3-hydroxy-3-acetonyloxindole
C_16_H_18_N_2_O_6_	[M−H]^−^	indole-3-cyano-6-o-glucoside
C_16_H_18_N_2_O_5_S	[M−H]^−^	indole-3-cyano-2-o-glucoside
C_10_H_8_N_2_	[M+H]^+^	3-indoleacetonitrile
C_9_H_7_NO_2_	[M−H]^−^	indole-3-carboxylic acid

## Data Availability

All data are available within the article or its Appendix A.

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
