# Peer review of "The Aqueous Extract of Brassica oleracea L. Exerts Phytotoxicity by Modulating H2O2 and O2 Levels, Antioxidant Enzyme Activity and Phytohormone Levels"

_plants, 2023, doi:10.3390/plants12173086_

Round 1

Reviewer 1 Report

The experiment conducted by the authors broadens the knowledge on the “The Aqueous Extract of Brassica oleracea L. Exerts Allelopathic Effects by Modulating ROS Levels, Antioxidant Enzyme Activity and Phytohormone Level”. There are a lot of comments that should be taken into account by authors, which I believe are significant and important aspects that need to be thoroughly addressed in authors revision.

The main concern is:

Abstract:

(1) The abstract is very poorly constructed. For example, the basic experiment that was done to conclude their conclusion is not clearly revealed in the abstract. This part needs to be completely re-written, presenting the results interesting to the readers.

(2) The authors should follow abbreviation rule – first define a term, and then use the abbreviated form. e.g. L18: H2O2 and O2 should be defined.

Introduction:

(3) Again, abbreviations should be defined in first mention, please revise this issue in the whole ms!

(4) I am wondering many vital and recent papers are missing. Please illustrate by more details the impact of using extracts on the measured parameters.

(5) Lines 48-49 add a reference.

(6) Lines 56-57: this sentence should be rephrased.

Results:

(7) Write the headings and sub-headings in sentence case throughout the MS.

(8) Please, a systematic description of your results must be done, highlighting for the reader observations that are most relevant to the topic under investigation and it can be used in the discussion later. It is recommended to highlight more important results.

(9) In figure (1) caption. Authors mentioned to (*P < 0.05, **P < 0.01). Actually, I cannot find them in the figure. Please revise this issue in the whole figures!

(10) Figures (2) and (3) resolution should be raised. Please improve the quality of the figures. 

Discussion:

(11) Please, a deeper scientific interpretation of your findings in the discussion section is strongly suggested. The novelty and implication of your study should also be highlighted throughout this section.

Material and methods:

(12) In material and methods, please cite all described methods because these methods are not your own.

Conclusion:

(13) It should be concise and should be conclusively explained in one paragraph. It should also provide guidelines for future research.

References

(14) Write the citations and references according to journal format.

Linguistic quality:

(15) The language used in the manuscript needs improvement in sentence structure and writing style. The current sentence structure is lacking in proficiency. A professional English editing service must edit your manuscript.

 Extensive editing of English language required

Reviewer 2 Report

The manuscript entitled “The Aqueous Extract of Brassica oleracea L. Exerts Allelopathic 2 Effects by Modulating ROS Levels, Antioxidant Enzyme Activity and Phytohormone Level” reports information on the phytotoxic effect of aqueous extract of Brassica oleracea on the crop Panicum miliaceum to find bioherbicides. The objective of this work was to find potentialities to use Brassica oleracea extract to control weeds.

The topic is interesting. However, and after reading the text, I decide to stop reviewing in the Results section. The reason is that I found serious drawbacks that need to be addressed. 

The main concerns are: 

1- Authors did not evaluate allelopathy but phytotoxicity. Allelopathy can only invoked when plant allelochemical are obtained in a natural way and natural concentrations (e.g. rainfall, root exudates, etc.). However, Brassica phytotoxins were obtained by grinding plant material, therefore releasing phytotoxins that could not be present in natural conditions. Also, they used an arbitrary concentration of phytotoxins. 

2- If the objective of this work was evaluate the herbicide potential of Brassica oleracea extract, why did they use a crop as a target species? Authors, must justify why they used a crop such as P. miliaceum instead of known-problematic weeds. 

3- Material and Methods section was poorly described, with entire procedures and a lot of data missing. Someone cannot repeat this study with the material and methods as described. 

4-Results section is confusing. I found a great number of errors and discrepancies between results description and figures, which lead to a misinterpretation of results.

I explained these comments and other minor comments in more detail on the attached pdf file as pink, red and blue marks and pop-up notes.

I am not qualified to assess the quality of English in this paper

Reviewer 3 Report

This paper did many works and data obtaioned have value to be reported. There are several points better to change. Ss for this, I add my comments directly on the paper. As for Fig.1, it is better to change line graph and estimate EC50 value, for example, Fig1-a, about 0.16 g/mL. As for the contribution of plant hormons, I can not follow that you explained that the concentration of hormons identified in the extract can fully explain the results. Please explain this in the text or by Figures. 
